# The Assessment of the Efficacy, Safety, and Challenges of Ketogenic Diet Therapy in Children with Epilepsy: The First Experience of a Single Center

**DOI:** 10.3390/medicina60060919

**Published:** 2024-05-31

**Authors:** Jurgita Karandienė, Milda Endzinienė, Karolina Liaušienė, Giedrė Jurkevičienė

**Affiliations:** 1Neurology Department, Hospital of Lithuanian University of Health Sciences Kauno Klinikos, 50161 Kaunas, Lithuania; milda.endziniene@lsmu.lt; 2Neurology Department, Lithuanian University of Health Sciences, 50162 Kaunas, Lithuania; giedre.jurkeviciene@lsmu.lt; 3Independent Researcher, 53203 Kaunas, Lithuania; karolina.liausiene@outlook.com

**Keywords:** ketogenic diet, epilepsy, seizure, children, challenges, drug-resistant epilepsy, patient compliance

## Abstract

*Background and Objectives*: Ketogenic diet therapy (KDT) has been used as a non-pharmacological treatment for childhood refractory epilepsy. Its efficacy and safety have been described in numerous studies and reviews. However, there have been fewer studies evaluating the challenges experienced by patients and their family members when starting KDT. When implementing a new treatment method, challenges arise for both the healthcare professionals and patients, making it important to summarize the initial results and compare them with the experiences of other centers. To analyze and evaluate the efficacy and safety of KDT in children with epilepsy, as well as to consider the challenges faced by their parents/caregivers. *Materials and Methods*: A retrospective analysis of patients’ data (N = 30) and an analysis of the completed questionnaires of the parents/caregivers (N = 22) occurred. *Results*: In the study group, 66.7% of the patients had a >50% decrease in seizure frequency, and 2/3 of them had a >90% decrease in seizure frequency or were seizure-free, which enabled reducing the anti-seizure medications in 36.4% of the patients, as well as reducing the hospital visits. Cognitive improvement and better alertness were subjectively reported by 59.1% of the parents/caregivers. No dangerous long-term adverse effects of KDT have been observed in the study group. The patients with generalized epilepsy experienced significantly more adverse events. Most of the adverse effects of KDT were related to the digestive system, but usually they were temporary and controllable. The challenges of the parents/caregivers were mostly related to social life issues and financial difficulties; the medical-related challenges were minimal. *Conclusions*: KDT is an effective and safe treatment option for children with drug-resistant epilepsy, and the challenges faced by families are resolvable. In order to ensure effective KDT, a multidisciplinary team is required. This would ensure smooth and comprehensive care and the timely resolution of emerging problems. The cooperation of the families undergoing KDT is also important, enabling them to share their experiences.

## 1. Introduction

The ketogenic diet is a high-fat, low-carbohydrate, and adequate-protein diet that has been used as a therapeutic option for patients with drug-resistant epilepsy. Numerous studies have shown that ketogenic diet therapy (KDT) can be effective in reducing the seizure frequency and improving the seizure control in children and adults who do not respond well to anti-seizure medications (ASMs), have difficulty tolerating them, or when epilepsy surgery is impossible [1,2,3,4,5]. For some conditions, such as glucose transport type-1 (GLUT-1) deficiency syndrome and pyruvate dehydrogenase complex (PDHC) deficiency, KDT is the treatment of choice [1,3,6], but KDT can also be effective in certain types of epilepsy, such as Dravet syndrome [7,8,9], Lennox–Gastaut syndrome [3,10,11,12], tuberous sclerosis complex [3,13,14], infantile spasms, etc. [1,3,15]. Moreover, a ketogenic diet is recommended as a new adjunctive treatment during critical care for the resolution of acute status epilepticus and related disorders, such as new-onset refractory status epilepticus (NORSE) and febrile infection-related epilepsy syndrome (FIRES), when the traditional ASMs and anesthetic agents fail [16,17,18].

The safety of undertaking a ketogenic diet depends on various factors, including the individual’s medical history, overall health, and specific dietary needs. The most commonly described complication of introducing KDT is hypoglycemia. Dehydration is more common when KDT is introduced after a few days of fasting. Additionally, gastrointestinal symptoms often occur at the beginning of KDT: vomiting, nausea, diarrhea, and abdominal pain. These side effects are usually short-lived and can be corrected by reviewing the dietary plan. Some serious complications that sometimes occur include kidney stones and pancreatitis [18,19,20].

The implementation of the ketogenic diet can often be a challenge for the family. When applying the ketogenic diet, the parents/caregivers not only need to learn how to accurately calculate the nutrient content of meals and discover new recipes adapted to the diet but also need to review the dietary habits of the entire family. This can create difficulties in the social aspects of family life [21,22,23].

While there are international recommendations for the use of KDT [3,6], introducing a new treatment method poses challenges for both the healthcare professionals and patients, as well as their families. Close collaboration between the patients and the KDT team can help to mitigate these challenges and achieve better outcomes. However, there are significantly more studies evaluating the efficacy and safety of KDT in treating drug-resistant epilepsy but far fewer that describe the challenges faced by both families [23,24] and healthcare professionals.

Since 2019, KDT has been implemented and applied in the Child Neurology Department at the Hospital of the Lithuanian University of Health Sciences Kauno klinikos (Kauno klinikos) for the treatment of children with drug-resistant epilepsy. Until then, the application of KDT in Lithuania was limited to individual cases, primarily initiated by parents/caregivers, without a regular monitoring plan. Additionally, there have been no publications regarding the use of this treatment method in Lithuania. As the non-pharmacological treatment options for drug-resistant epilepsy have been expanded at Kauno klinikos, it is crucial to review the initial results of KDT and compare them with the data published by other centers.

The aim of the study was to analyze and evaluate the efficacy and safety of KDT in children with epilepsy, as well as the challenges faced by their parents/caregivers.

## 2. Materials and Methods

The study included pediatric patients with epilepsy who were treated with the ketogenic diet as inpatients/outpatients at the Child Neurology Department of Kauno klinikos from 1 April 2019 to 1 October 2022, and whose parents/caregivers agreed to participate in the study and sign the informed consent form. Before starting KDT, patients are consulted by a pediatric neurologist (evaluation of epilepsy form, seizure type and frequency, detecting indications and contraindications for KDT, neurological examination, and correction of treatment with ASMs), a dietitian (assessment of nutritional status and feeding pathway, meal preferences, allergies, and calculation of calories and diet ratio), and comprehensive laboratory (complete blood count, blood biochemistry, urine analysis, ASMs blood levels, serum acylcarnitine profile, serum amino acids profile, and urine organic acids profile) and instrumental tests (electroencephalogram, brain magnetic resonance imaging, electrocardiogram, and abdominal and renal ultrasonography) are performed. The classic ketogenic diet was applied to our patients by gradually increasing the ketogenic ratio to 3:1 or 4:1, using specialized formulas as needed. All patients began treatment as inpatients. Some sources of information were provided to parents/caregivers and children: booklets in the Lithuanian language and a video for children about KDT with Lithuanian subtitles (https://www.youtube.com/watch?v=olZCljOSeZ8, accessed on 12 May 2024). Also, a special email address for consultations regarding KDT has been created (ketogenine.dieta@kaunoklinikos.lt), as well as an online forum (Facebook group “Gydanti keto mityba vaikams”). Parents/caregivers and children are trained in calculating meal plans and measuring ketosis and glycemia. Instructional materials and special monitoring charts have been created and provided for home use. All patients reached and maintained therapeutic ketosis (blood ketone level 2–5 mmol/L). According to international recommendations [1,3,25,26], during the follow-up period, regular outpatient visits were conducted—every 3 months during the first year of KDT, and then every 6 months. During these visits, the efficacy and adverse effects of KDT were assessed, blood and urine tests were performed, dietary plans were adjusted as needed, and necessary supplements were prescribed. Therefore, after the first 3 months of KDT, a decision is made whether to continue or discontinue the diet.

Data were collected by using two data sources: (1) objective demographic and clinical information was obtained from all available medical charts, and (2) subjective parent/caregivers’ opinions on the positive or negative effects of KDT and the challenges they had experienced were collected from questionnaires filled out by the study parents/caregivers. The data from both sources were analyzed in terms of three main aspects: (1) efficacy, (2) safety, and (3) challenges.

KDT is considered effective when the seizure frequency is reduced more than 50%, categorized into 3 groups: seizure reduction > 50%, seizure reduction > 90%, and seizure-free. To evaluate the safety of KDT, adverse effects were analyzed: nausea/vomiting, hunger, constipation, drowsiness, mood changes, hypoglycemia, hyperketosis, weight loss, and weight gain. It is important to note that we not only assessed the presence of adverse effects but also asked parents/caregivers to evaluate whether these adverse effects had temporary or long-term impacts. Challenges encountered by families during KDT implementation were assessed in focus groups: accessibility of information about KDT, dietary changes, financial and social aspects, and accessibility of medical services.

### Statistical Analysis

The statistical data analysis was performed using the IBM SPSS Statistics 23.0 software package. The distributions of quantitative variables were assessed visually and using the Shapiro–Wilk test. KDT was considered effective when the frequency of seizures decreased by more than 50%. For examining differences in KDT efficacy between genders, epilepsy types, and special formula consumption, the non-parametric chi-squared (χ^2^) test was used. Adverse effects were encoded into scores, with a score of 2 indicating a long-term impact on parent’s satisfaction, a score of 1 indicating a temporary impact, and a score of 0 indicating no observed adverse effects, while the maximum score of 16 would indicate the presence of long-term adverse effects in all areas examined. The sum of scores expressing the adverse effect was calculated for each participant. To compare the duration of KDT and the sum of scores for adverse KDT effects between two groups of KDT efficacy, as well as when assessing the differences in the sum of scores for adverse KDT effects between genders and different epilepsy forms, the non-parametric Mann–Whitney U test was used. To calculate the differences in the sum of scores for adverse KDT effects among different epilepsy etiological types, the non-parametric Kruskal–Wallis test was used. The Spearman’s rank correlation coefficient was used to analyze the relationship between the duration of KDT and the quantity of adverse effects. A significance level of 0.05 was used when testing statistical hypotheses.

## 3. Results

During the study, the demographic and clinical data of 30 patients were analyzed (Table 1). Data from eight patient questionnaires were not included in the analysis of the parent/caregiver questionnaires due to data unavailability: three due to patient deaths (questionnaires were not sent for ethical reasons) and five due to incomplete parent/caregiver questionnaires. Therefore, twenty-two completed questionnaires from the parents/caregivers were analyzed (Figure 1). In accordance with this, the size of the group described is provided in brackets.

Within the study period, 13 (43.3%) continued the KDT treatment. Among the remaining seven participants (23.3%), KDT was discontinued due to lack of efficacy, five participants (16.7%) discontinued due to intolerance, three participants (10%) died, and two participants (6.7%) discontinued due to a lack of motivation (Figure 1). The average duration from the start of the treatment until discontinuation was 9.5 ± 7.9 months.

### 3.1. Efficacy of KDT in Children with Epilepsy

KDT was found to be effective (reducing seizure frequency > 50%) in twenty participants (66.7%, N = 30), with four participants (13.3%) becoming completely seizure-free (Table 2).

When the parents/caregivers were asked about the positive effects of KDT, they (N = 22) reported that twelve (54.5%) experienced a reduction in seizure frequency, and four (18.2%) became seizure-free. Furthermore, eight (36.4%) of the patients experienced milder/shorter seizures. The parents also noticed a subjective positive effect of KDT on psychosocial well-being: thirteen (59.1%) of the patients were more alert, nine (40.9%) experienced improved sleep, and eleven (50%) showed developmental improvements. The parents also noted positive effects related to epilepsy treatment: four (18.2%) of the patients were able to reduce the dosage of ASMs, eight (36.4%) were able to discontinue at least one medication, and ten (45.5%) had fewer visits to healthcare facilities. However, in five cases (22.7%), no positive effects were reported by the parents/caregivers.

### 3.2. Safety of KDT in Children with Epilepsy

Table 3 describes the adverse effects of KDT in the study group (N = 22). The average sum of scores for adverse effects (0 indicated no observed burden of adverse events, while the maximum score of 16 would indicate the presence of long-term adverse effects in all the areas examined) was 4.6 ± 2.4 (range 0–10). Comparing the sum of scores for adverse effects between the two effectiveness groups (with or without effect), the average ranks of the sum of scores did not significantly differ between these groups (Mann–Whitney U test, U = 30, z = −1.4, *p* = 0.176), although a difference was observed (medians of 4 and 6, respectively). The average ranks of the sum of scores for adverse effects did not significantly differ between the boys and girls (Mann–Whitney U test, U = 33.5, z = −1.7, *p* = 0.089) (medians of 6 and 4, respectively). It was found that the patients with generalized epilepsy experienced significantly more adverse effects as compared to the patients with focal epilepsy (Mann–Whitney U test, U = 19.5, z = −2.1, *p* = 0.032) (medians of 5.5 and 3.5, respectively). The average ranks of the sum of scores for adverse effects did not significantly differ among the different types of epilepsy (Kruskal–Wallis test, H (2) = 4.1, *p* = 0.396). The analysis of the influence of KD duration on the occurrence of adverse effects revealed a negative but non-significant correlation (Spearman’s correlation, r = −0.2, *p* = 0.317).

### 3.3. Challenges of Introducing KDT

In the survey (N = 22), the parents/caregivers of twelve children (54.5%) indicated that KDT was continued, seven (31.8%) had discontinued the diet, and three (13.6%) reported adopting a modified version of the ketogenic diet. Among the parents/caregivers of children who no longer followed KDT, five (22.7%) noted that this was because they did not observe the expected effects, one (4.5%) reported that the child could not tolerate the diet, one (4.5%) mentioned that the child refused to eat the required food and violated the diet, two (9.1%) reported significant adverse effects, and one (4.5%) was suggested another more effective treatment option. The decision to discontinue KDT was unilaterally made by the parents/caregivers of one child, while, in eight cases (27.3%), the decision was made together with the healthcare professionals in charge of the KDT at the Kauno klinikos.

The majority of the parents/caregivers of the participants, 16 (72.7%), learned about KDT from a pediatric neurologist. Some parents, six (27.3%), learned about the diet from the media and/or online portals, four (18.2%) from online forums and/or social media groups, three (13.6%) from other parents, and two (9.1%) from other specialists. Eighteen participants (81.8%) were orally fed before starting the ketogenic diet, three (13.6%) were fed through a gastrostomy, and one (4.5%) had a mixed feeding method.

Figure 2 describes the challenges faced by the parents/caregivers during the initiation of ketogenic diet therapy.

The parents/caregivers mostly lacked information on how to calculate the nutrient requirements and caloric content of food, eight (36.4%), how to prepare meals, five (22.7%), and how to handle special situations, such as illness, trauma, anesthesia, or surgery, five (22.7%). In the survey, two (9.1%) indicated that they faced a specific challenge. In addition, eighteen (81,8%) of the parents/caregivers indicated a lack of information regarding the use of dietary supplements, one (4.5%) on how to manage high or low levels of ketones and glucose in the blood, while eight (36.4%) stated that they did not lack any information at all.

Regarding additional information, 21 (95.5%) of the parents/caregivers were seeking it online, and 19 (86.4%) obtained it from specialized forums/groups and during medical consultations. Social media platforms as information sources were mentioned by 17 (77.3%).

When asked about ways in which parents/caregivers contribute to helping themselves and other parents, it was found that almost all of them, 19 (90.9%), responded to questions when personally asked by others. Additionally, half of them, 11 (50%), engaged in personal communication with one or more families of children undergoing KDT. Eight (36.4%) parents/caregivers shared supplements and resources, while five (22.7%) shared their own discovered recipes. They also initiated conversations and asked questions in the Facebook group. Only two (9.1%) parents indicated that they did not participate in these activities.

## 4. Discussion

In our study, we observed that KDT was effective in 66.7% of the patients, with exceptionally good results reported in 44.6% of the cases, which is consistent with the existing literature [20,23,27,28,29,30]. More than half (59%) of the surveyed patients’ parents/caregivers also expressed a positive response of KDT in terms of epilepsy seizure control, such as a decrease in seizure frequency, milder and shorter seizures, or seizure-free status. This is particularly important as the studied patients had intractable epilepsy and had tried various combinations of ASMs. Furthermore, over half of the parents/caregivers noticed an improvement in cognitive functions, including increased alertness, improved communication, developmental progress, and better sleep [31]. Almost all the participating patients had documented cognitive deficits, which, when accompanied by recurrent, frequent, and prolonged seizures, worsen developmental progress and hinder communication and socialization. Therefore, the improvement in these functions observed by the parents when implementing KDT is crucial for evaluating the treatment effectiveness and the long-term development, family interaction, and socialization of the patients [24].

Additionally, 36% of the parents/caregivers indicated in the surveys that, in addition to the positive effects of KDT, they were able to reduce the dosage of ASMs or discontinue at least one medication. This is significant because reducing the doses and quantity of ASMs reduces the potential side effects and enhances patients’ alertness [1,29]. It also has a positive impact on the cost of epilepsy treatment [32].

It is important to mention that 36.4% of the parents/caregivers reported a reduction in visits to healthcare facilities. It can be assumed that, with a lower seizure frequency and fewer prolonged seizures, there is a reduced need to seek emergency care or attend outpatient visits. A lower frequency of visits to healthcare facilities can have a positive impact on the family’s social life and reduce the transportation expenses. Fewer visits to healthcare facilities, along with a lower quantity of prescribed ASMs, also reduce the societal costs of patient treatment [33].

The patients for whom KDT was effective continued the diet for a longer duration. In our study, we observed an unexpected significant difference in the efficacy of KDT between the boys and girls, with the girls experiencing greater efficacy. There is no corresponding comparison found in the literature regarding this gender-based difference in efficacy. The reason for the observed association between gender and efficacy might be due to the small study sample. This observation could be further explored in future prospective studies.

A statistically reliable difference in KDT efficacy between generalized and focal epilepsy has not been found, although there is a tendency for KDT to be slightly more effective for generalized epilepsy. To assess this more accurately, a larger sample size would be necessary. There are individual articles in the literature suggesting that KDT is more effective for patients with generalized epilepsy [34], which could be a subject for future research.

The results of our study indicate that the patients in the research group achieved and maintained therapeutic ketosis, which demonstrates that KDT was implemented and followed correctly. Therefore, our presented results can be considered reliable [35]. In order to assess the associations between the KDT adherence and achievable outcomes, specialized questionnaires were developed and implemented into practice [36,37].

Our study shows that KDT was relatively safe, with few significant adverse events observed. The most commonly reported adverse effects by the parents/caregivers were related to the digestive system: nausea, vomiting, and constipation. Long-term constipation was reported by nearly one-third of the patients, as also indicated in the literature [19]. This could be associated not only with KDT but also with the fact that children with drug-resistant epilepsy and comorbid movement disability are prone to constipation. Nausea and vomiting were temporary phenomena, which are also reported in the literature as more common at the beginning of KDT due to the physiological changes in the gastrointestinal tract and as an expression of patient resistance to the new diet [19]. More than half of the patients did not indicate hunger or weight fluctuations as an adverse effect. According to the testimonies of two parents, weight gain was a positive effect due to the slow weight gain of the patient prior to KDT. The parents/caregivers noted encountering clinically significant hyperketosis and hypoglycemia. These conditions can be managed by taking additional measures based on hypoglycemia or hyperketonemia algorithms, which are provided to parents by healthcare professionals responsible for KDT implementation.

The adverse effects of KDT reported by patients’ parents/caregivers were quantified on a score scale. The average score was 4.6 (from a maximum of 16). Therefore, in our group of patients, the overall burden of adverse events did not exceed one-third of the maximum possible score. Thus, we can conclude that the burden of adverse events in our patient group was relatively low, especially considering that some of these adverse events were temporary and manageable. It was found that the patients with generalized epilepsy experienced significantly more adverse events than those with focal epilepsy. It is possible that the patients with generalized epilepsy had a more severe overall condition, and a larger sample size and longer follow-up would be needed for a more precise assessment.

KDT is a long-term change that requires the efforts and dedication of the entire family, posing additional challenges for them. The parents/caregivers most commonly identified the difficulties and costs of procuring appropriate ketogenic diet products as the largest and long-lasting challenge. Calculating the food composition, meal preparation, and the regular testing of ketones and glucose at home typically presented temporary challenges that could be overcome with time. Even 72.7% of the parents/caregivers noted that KDT caused more difficulties in eating away from home after beginning KDT. This can further worsen the social life of families with children with epilepsy. Nearly one-third of the patients’ parents/caregivers reported difficulties in family relationships after implementing KDT. Only a small number of parents/caregivers (1 in 22) indicated challenges related to regular visits to healthcare facilities, obtaining specialist consultations, or receiving information in their native (Lithuanian) language. Therefore, the majority of the challenges arise from social and economic issues rather than problems related to the provision and accessibility of medical services.

About half of the participants continued with KDT. The remaining participants discontinued the diet for various reasons: lack of efficacy, intolerance, child’s refusal to eat, or other proposed treatment options. The decision to discontinue KDT unilaterally was made only by 3.3% of the children’s parents/guardians. Most of the parents/caregivers discussed the decision to discontinue the treatment with the child’s doctor.

According to the survey of the parents/caregivers, a portion of them expressed a lack of information on how to calculate the daily nutrient requirements and caloric intake, prepare suitable meals for the diet, and manage special situations, such as illness, trauma, anesthesia, or surgery. Most of the parents/caregivers sought the missing information online, in specialized forums/groups. It is encouraging that 72.7% of the parents/caregivers obtained the necessary information during doctor consultations. The main amount of educational material was being prepared when KDT was already implemented at the Kauno klinikos, so the parents/caregivers of the initial patients had less information available. It is important to note that the informational resources were prepared in collaboration with the parents/caregivers, discussing and clarifying what information in what format was most needed. Of course, it is worth considering the need for more educational materials about KDT and organizing more events dedicated to it. More attention should also be afforded to educational activities involving not only families implementing KDT but also general practitioners and pediatricians providing children’s healthcare services.

Almost all the patients’ parents/caregivers were collaborative and were inclined to personally answer other parents’ questions about KDT, with half of the respondents stating that they personally interacted with one or more families of children undergoing KDT. This highlights the importance of a self-help community that can assist patients’ families in navigating the daily challenges [24].

Changes in daily dietary habits can cause tension not only for the patients themselves but also for their family members [23]. Therefore, close collaboration between families, healthcare specialists, easily accessible information, and overcoming the difficulties encountered in implementing KDT are of great importance. Based on our surveys of the parents/caregivers and clinical data obtained by our team, the positive effects of KDT on seizure control and the cognitive development of the child outweigh the challenges that may arise. So far, individual studies have been conducted, showing a tendency for the efficacy of KDT to outweigh the challenges faced by families [24], but more research and specialized questionnaires are needed to evaluate the impact of KDT on quality of life [38].

To ensure the smooth implementation, application, and patient care of KDT, a competent multidisciplinary team is necessary. This team should consist of a pediatric neurologist, dietitian, specially trained nurse, and psychologist, who can dedicate sufficient time for consultations and addressing any arising questions. This would ensure comprehensive and seamless patient care and the timely resolution of any issues that may arise. To further investigate the factors related to the safety and efficacy of KDT, as well as the changes in the laboratory blood indicators during its application, conducting a longer-term prospective study would be beneficial. Additionally, collaboration among the families implementing KDT is crucial, not only in working with healthcare professionals but also in fostering communication, sharing experiences, and organizing joint events.

## 5. Conclusions

In conclusion, KDT has been evaluated as an effective and safe therapeutic option for children with epilepsy, particularly those who do not respond well to medications. It can help to reduce seizure frequency, improve seizure control, increase parents’ satisfaction, and reduce the medical costs. However, it requires careful implementation, monitoring, and medical supervision to ensure its efficacy and safety. Each child’s individual needs and medical history should be considered when considering KDT as a treatment option for epilepsy.

## 6. Limitations of the Study

The limitations of our study include a small sample size and a short monitoring period. In order to more objectively assess the changes in the quality of life of patients and their families, it is recommended to use specialized questionnaires that are adapted to the specific challenges that arise during the ketogenic diet and for evaluating its effects.

## Figures and Tables

**Figure 1 medicina-60-00919-f001:**
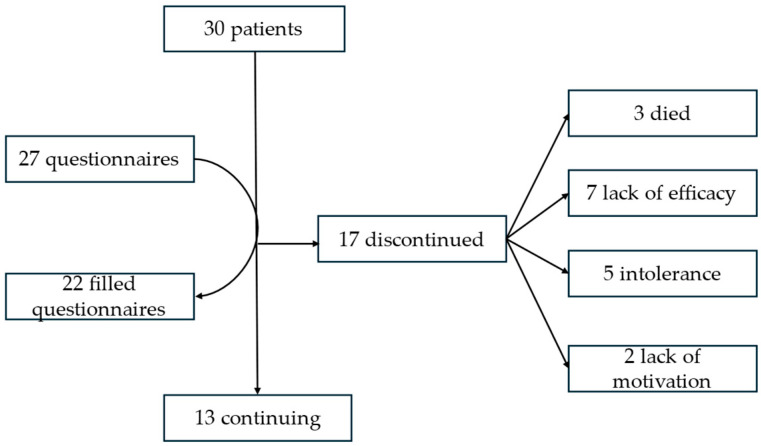
Flow diagram of the study group.

**Figure 2 medicina-60-00919-f002:**
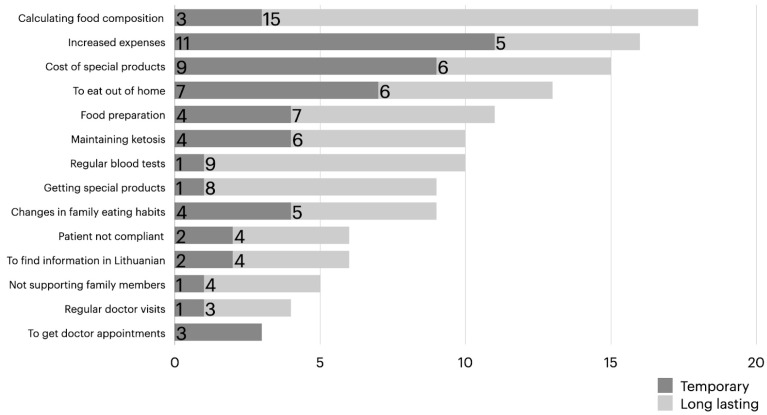
The challenges experienced by parents/caregivers (N = 22) when implementing ketogenic diet therapy.

**Table 1 medicina-60-00919-t001:** Demographic and clinical data of the study group (N = 30).

	Total	Boys	Girls
Gender, n (%)	30 (100)	16 (53.3)	14 (46.7)
Epilepsy duration before introducing KDT; months	48 ± 34.1	42.6 ± 33.5	54.3 ± 34.9
Age of KDT initiation; years	6.6 ± 4.6	5.8 ± 3.2	7.4 ± 5.7
Average KDT duration; months	15.6 ± 12.5	13.8 ± 14	17.8 ± 10.8
Continuing KDT, n (%)	13 (43.3)	5 (31.3)	8 (57.1)
Type of epilepsy, n (%):	30 (100):	-	-
- focal	10 (33.3)	-	-
- generalized	20 (66.7)	-	-
Number of ASMs before KDT	6.6 ± 2.2	-	-
Number of ASMs at the start of KDT	3.3 ± 1	-	-

KDT—ketogenic diet therapy. ASMs—anti-seizure medications.

**Table 2 medicina-60-00919-t002:** Efficacy of ketogenic diet therapy in the study group.

	KDT Effective	KDT Not Effective	*p* Value
Overall efficacy, n (%)	20 (66.7)	10 (33.3)	0.0201
- seizure reduction > 50%	6 (20.1)	-	-
- seizure reduction > 90%	10 (33.3)	-	-
- seizure free	4 (13.3)	-	-
Gender, n (%):			
- boys	8 (50) *	8 (50)	* 0.038
- girls	12 (85.7) *	2 (14.3)	
Epilepsy type, n (%):			
- focal	8 (80) **	2 (20)	** 0.273
- generalized	12 (60) **	8 (40)	
Average KDT duration, months	20.1 ± 11.1	6.3 ± 2.7	0.000
Using special formula, n(%)			
- used	12 (80) ***	3 (20)	*** 0.121
- not used	8 (53.3) ***	7 (46.7)	

* Efficacy between genders. ** Efficacy between epilepsy type. *** Efficacy based on special formula use. KDT—ketogenic diet therapy.

**Table 3 medicina-60-00919-t003:** The adverse effects of ketogenic diet therapy for children with epilepsy, as indicated by parents/caregivers (N = 22).

Adverse Effect	Long-Term Impact	Temporary Impact	No Adverse Effect
	n (%)		
Nausea/vomiting	3 (13.6)	12 (54.5)	7 (31.8)
Hunger	2 (9.1)	3 (13.6)	17 (77.3)
Constipation	7 (31.8)	5 (22.7)	10 (45.5)
Drowsiness	2 (9.1)	8 (36.4)	12 (54.5)
Mood changes	5 (22.7)	8 (36.4)	14 (63.6)
Hypoglycemia	2 (9.1)	3 (13.6)	17 (77.3)
Hyperketosis *	2 (9.1)	6 (27.3)	14 (63.6)
Weight loss	2 (9.1)	3 (13.6)	17 (77.3)
Weight gain	3 (13.6)	3 (13.6)	16 (72.7)

* Hyperketosis—undesirable elevation of blood ketone levels > 6 mmol/L, requiring adjustment.

## Data Availability

The data that support the findings of this study are available from the corresponding author upon reasonable request.

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
