# Peer review of "The Assessment of the Efficacy, Safety, and Challenges of Ketogenic Diet Therapy in Children with Epilepsy: The First Experience of a Single Center"

_medicina, 2024, doi:10.3390/medicina60060919_

Round 1

Reviewer 1 Report

Comments and Suggestions for Authors

This is an article about the use of Ketogenic Diet Therapy (KDT) in Lithuania. Please find below some suggestions to improve the quality of the paper:

1.       Please make it clear that it is all about Lithuania. Otherwise seems it is nothing special or new.

2.       Please use the recently published definition of KDT DOI: 10.1016/j.numecd.2024.01.014

3.       Lines 72 and 74: there are extra spaces between some words, please check format;

4.       Line 88: instead of substances, it is better macronutrients.

5.       Line 90: please insert how many days or weeks took to reach the ketonemia and the mean +/- SD of the patient’s ketonemia.

6.       Line 91: what is limited time? Please be more specific;

7.       Line 95: Please make it clearer the which questionnaires, if they are validated or not, etc: “subjective parent/caregivers’ opinions on the positive or negative effects of the KDT and the challenges they had experienced were collected from questionnaires filled out by the study parents/caregivers”

8.       Line 98: These are results, not methods.

9.       Figure 1: it is not clear in the flowchart all these information: “1) efficacy, 2) safety, 3) challenges.”

10.   Line 112: it is not clear “were assessed in several groups”: focus groups?

11.   Line 121: it is not mentioned the questionnaire used to assess “quality of life”

12.   Table 1: Please give the real P number in all tables, also provide a comparison between the drop-out and remaining sample (n=22)

13.   It is necessary a better description of the prescribed diet (calories, macro, and micronutrients), ratio progression, supplementation, initiation, in or out-patient initiation, etc.

14.   Please also comment during the discussion if these results are correlated with the treatment choice (classical KDT?) or intervention methods or better adherence of patients or team effort etc….

Author Response

Dear Reviewer,

Thank you for your attention to my manuscript and your suggestions on how to improve it.

There is my few answers to your notes:

5. It was a retrospective study, and we didn’t check this point.

7. This questionnaire wasn’t validated but it was approved by the Ethics Committee for Biomedical Research at the Lithuanian University of Health Sciences.

9. This was some text formatting mistake and I corrected it.

11. We didn’t use a validated questionnaire, this was parent’s opinion.

12. We didn’t compare N=30 and N=22 groups because there weren't different groups, patients were the same, but N=30 was an objective analysis of clinical data and N=22 was an anonymous questionnaire of the parents. Also I want to apologize for the left mystipe in the previous variant of the manuscript in Table 2 – overall efficacy p value is 0.0201.

Please find attached my revised manuscript.

Kind regards,

Jurgita Karandiene

Reviewer 2 Report

Comments and Suggestions for Authors

file attached.

Author Response

Dear Reviewer,

Thank you for your attention to my manuscript and your suggestions on how to improve it.

There is my few answers to your notes:

  1. I am sorry, but I didn’t understand what I should correct here. List of adverse effects was of 9 terms, not 23 (Table 3). We wanted to show that despite the various adverse effects there wasn't a high amount of them, most were short-living, and the burden of adverse effects wasn’t very high for families.
  2. It was a retrospective study; we didn’t calculate sample size before the study.
  3. and 12. We didn’t use validated scales for cognitive function, this was a subjective parent’s evaluation.
  4. It is written in the end (lines 379-381) according to manuscript sample
  5. This isn’t interventional study, this was retrospective study of patient’s data
  6. I apologize for the left mistype in my Table 2. We looked multiple times and didn’t notice this… Overal efficacy p value is 0.0201.

Please find attached my revised manuscript.

Kind regards,

Jurgita Karandiene

Round 2

Reviewer 2 Report

Comments and Suggestions for Authors

Improved